# Transcriptome-Based Weighted Correlation Network Analysis of Maize Leaf Angle Regulation by Exogenous Brassinosteroid

**Xiangzhuo Ji** [1,2,3], **Qiaohong Gao** [1,2,3], **Zelong Zhuang** [1,2,3], **Yinxia Wang** [1,2,3], **Yunfang Zhang** [1,2,3] **and Yunling Peng** [1,2,3,*]

1    College of Agronomy, Gansu Agricultural University, Lanzhou 730070, China
2    Gansu Provincial Key Lab of Aridland Crop Science, Gansu Agricultural University, Lanzhou 730070, China
3    Gansu Key Lab of Crop Improvement & Germplasm Enhancement, Lanzhou 730070, China
\*    Correspondence: pengyl@gsau.edu.cn; Tel.: +86-138-9323-8528

**Abstract:** Maize (*Zea mays* L.) leaf angle is an important characteristic affecting high-density planting, and it is also a central indicator for maize plant type selection to improve yield. Brassinosteroids (BRs) are a class of phytohormones that could modulate the growth and development of plant leaf angles. However, its functional mechanism remains unclear in maize. In this study, we used maize self-line B73 as material to analyze the transcriptome of leaf cushion after BR treatment at the seedling stage. Using seven concentrations of exogenous BR-treated maize B73 plants, the results show that the leaf angle and the cell length near the leaf pillow increased and then decreased with BR concentration increasing, and the 50 μM level was the best treatment. Analysis of 11,487 differences expressed genes (DEGs) found that genes related to cell volume were up-regulated, and the expression of genes related to the cell division was down-regulated. It is speculated that exogenous BR regulates the size of the maize leaf angle by regulating cell volume and cell division, and so we constructed a molecular mechanism model of maize response to exogenous BR. The molecular mechanism model of exogenous BR through weighted gene co-expression network analysis (WGCNA) DEGs, and two gene modules related to changes in maize leaf angle were identified. The results can provide a theoretical basis for determining the mechanism of exogenous BR-regulated maize.

**Keywords:** maize; leaf angle; BR; transcriptome analysis; WGCNA

## 1. Introduction

The maize leaf angle is the angle between the leaf and the main stem, which is the primary factor determining the plant type of maize [1]. Furthermore, it is highly correlated with maize architecture and yield, making it is one of the most important agronomic traits [2]. The upper leaves of maize with a smaller leaf angle, which is profitable for the middle leaves to capture light, thus enhance photosynthesis and then increase planting density and yield [3–5]. A compact plant type with erect leaves is preferred since it increases photosynthetic efficiency and nitrogen storage in the grain-filling stage [6]. The leaf lamina joint, which connects the leaf blade and sheath, is considered the most important tissue controlling the leaf angle. The degree of the leaf angle largely depends on cell division and expansion as well as cell wall composition at the joint [7]. Therefore, breeding maize have more erect leaves and have been an appropriate strategy for improving crop productivity.

The formation of leaf angle is influenced by a variety of factors. BR is an important hormone that regulates plant growth and development, notably in regulating leaf angle formation [8]. Furthermore, this regulatory effect requires interaction with many endogenous plant hormones [9,10]. Previous research has found that exogenous BR treatment stimulated the growth and development of the leaf angle in plants [9,11,12]. In addition, the leaf angle is jointly regulated by interactions between the BR signaling pathway and multiple other signaling pathways [11,13]. Studies found that brassinosteroids regulated

root quiescent center cell division and stem cell replenishment and is correlated with Ethylene Response Factor 115 (ERF115), which controls the transcription of genes linked to various biological processes related to growth and development [14]. Auxin/indole-3-acetic acid (Aux/IAA) proteins in auxin-signal trans-duction are involved in BR- mediated growth responses in a manner dependent on organ type [15]. BR regulates gene expression and plant development through receptor kinase signaling pathways [16]. Regulation of various functions by BR mostly branched at the two nodes of the BR primary signaling pathway: GSK3-like kinases and BZR1-like TFs. The GSK3-like kinase family plays central negative roles in BR signaling, and BZR1 is a key transcription factor in the BR signaling pathway. After dephosphorylation, it can directly regulate the expression of downstream genes in the BR signaling pathway [10]. GSK2 is one of the critical suppressors of BR signalling and targets transcription factor, and it can stabilize OVATE FAMILY PROTEIN 3 (OFP3) to suppress BR to reduce rice leaf angle of the top second leaf at the reproductive stage [17]. Mechanisms of BR-mediated leaf angle have been characterized in great detail in rice. Nevertheless, the role and mechanism of action of the BR hormone in maize is not extensive.

To date, genes involved in BR biosynthesis in maize, such as na2/ZmDWF1 (C-24 sterol reductase), brs1/ZmDWF4 (C-22 $\alpha$ hydroxylase), na1/Zmdet2(5$\alpha$ Reductase), and lil1/Zmbrd1 (Brassinosteroid-6-oxidase 1) were found could induce leaves upright [18–20]. Leaf angle QTL-mapped ZmRAV1 was found to regulate a brassinosteroid C-6 oxidase to alter endogenous brassinosteroid content and leaf angle [21]. The brassinosteroid biosynthesis gene, ZmD11, an ortholog of rice *DWARF11* (D11), could increase endogenous BR contents, shoot and root length, kernel length, kernel width, and kernel area [10]. In foxtail millet (*Setaria italica*), BR signaling was repressed by DROOPY LEAF1 (*DPY1*), which was cloned from maize, and could promote the proliferation of both clear cells and abaxial sclerenchyma cells in leaf veins and lignin deposition in leaf blades, thus supporting the upright leaf architecture [22]. Furthermore, using the method of transgene silencing to reduce the expression of BRASSIOTEROID INSENSITIVE 1 (*BRI1*) homologous genes in maize and the effect of BR signal transduction on maize plant type was revealed for the first time. BRI1 is the starting point of the entire BR signaling pathway, acts as a sensor for BR, and the cascade transmits BR signaling [23]. Studies also observed that RIP2 encode a RING finger E3 ligase protein that directly binds to ROLLED AND ERECT LEAF 1 (*REL1*), a key regulator of leaf morphogenesis, which is broadly involved in leaf tilt by coordinating BR signaling [24]. However, the underlying molecular mechanism of BR governing the control of maize leaf angle remains unclear.

In this study, we used different concentrations of exogenous BR to treat maize B73 plants on the third leaf of maize, which is fully expanded (V3 stage), and phenotypic traits, cytological analysis, transcriptome sequencing, combined with WGCNA analysis, to predict the molecular mechanism of BR regulating maize leaf angle. Our findings were used to explore important BR signal responsive genes in maize and provide a genetic resource for breeding.

## 2. Materials and Methods

### 2.1. Plant Materials and Material BR Treatment

Maize germplasm material B73 was provided by the Maize Breeding Research Group, College of Agronomy, Gansu Agricultural University. Use vermiculite to plant maize seeds in a flowerpot with a diameter of 10 cm, and cultivate in a light incubator at a relative humidity of 65% and 28 °C/25 °C 14 h day/10 h night, with a light intensity 8000 Lux. When at theV3 stage, different concentrations of exogenous BR (0, 0.1, 1, 10, 50, 100, and 150 $\mu$mol·L$^{-1}$, Solarbio, B8780, Beijing, China) are dissolve BR in 95% ethanol. Then, make mother liquor with distilled water and apply by spraying over leaves, which were replenished at 12 h intervals and repeated 2 times, with three replicates in each treatment. The leaf angle of each treatment was counted, and 2 cm in the lobe occipital region under 0, 1, and 50 $\mu$mol·L$^{-1}$ treatment were randomly collected and rapidly frozen in liquid nitrogen

for RNA-seq analysis, and abbreviated as B73-CK, B73-1BR, and B73-50BR, respectively. Each treatment collected three biological samples.

### 2.2. Ligular Region Longitudinal Structure Analysis

We selected the junction of the leaf auricle and leaf sheath on the third leaf of maize B73 to prepare paraffin sections. The experimental procedure are described by Chen et al. [25]. Longitudinal section cells of maize lamina joint were observed by LEICA DM500 (LEICA, Shanghai, China) optical microscope, and the length was measured by the ToupView (Microscope, Suzhou, China) camera system.

### 2.3. Total RNA Extractions, cDNA Library Construction, and Sequencing

The total RNA of 9 leaf pulvinus samples (B73-CK, B73-1BR and B73-50BR) were extracted using TRIzol reagent (Invitrogen, Carlsbad, CA, USA) according to the product manual. The total RNA concentration and quality were assessed using Agilent 2100 Bioanalyzer (Agilent Technologies, Santa Clara, CA, USA) and NanoDrop (IMPLEN, Westlake Village, CA, USA). The total RNA of each treated sample was taken for the construction of the RNA-seq library as described by Chen et al. [26]. Then, the cDNA libraries were sequenced with the BGISEQ-500 sequencing platform and 150 bp paired-end reads were generated. The entire process was commissioned and completed by BGI Genomics (Shenzhen, China). The original sequencing reads have been submitted to the SRA at NCBI (Accession number: PRJNA851970).

### 2.4. Transcriptome Analysis

Raw data in fastq format were firstly processed through in-house perl scripts. In this step, clean data were obtained by removing reads containing adapter, reads containing ploy-N, and low-quality reads from raw data. A summary of the RNA-Seq data was shown in Supplementary Table S1. Then the clean reads were aligned to the third version of the B73 maize reference genome (http://ftp.maizesequence.org, accessed on 17 April 2022) using HISAT software [27]. Gene expression levels were determined by RSEM software (BGI, Shenzhen, China) [28]. Differentially expressed genes (DEGs) between two treatments (B73-CK vs. B73-1BR, B73-CK vs. B73-50BR; B73-1BR vs. B73-50 BR) were identified, as described by Audic et al. [29]. The standard of DEGs screening was $p$-value $< 0.05$, false discovery rate (FDR) $\leq 0.001$, and $|\log2$ fold change$| \geq 1$. The functional annotations of the selected DEGs in the GO database were determined by WEGO software (BGI, Shenzhen, China) [30]. The metabolic pathways enriched in DEGs were determined by the Kyoto Encyclopedia of Genes and Genomes pathway database (https://www.genome.jp/kegg/kegg1.html, accessed on 16 April 2022).

### 2.5. Weighted Gene Co-Expression Network Analysis

Use R-package weighted gene co-expression network analysis (WGCNA) to construct a gene co-expression network from the expression data of B73-CK, B73-1BR, and B73-50BR [31]. Thresholds are set as follows: the average gene expression FPKM value is 1; the higher the similarity threshold fold of control module fusion, the lower the similarity required for fusion of 2 modules. Then, the fold is set to 0.5, and the minimum number of genes within a module is set to 30. Gene expression adjacency matrices were constructed and used to analyze the network topology. In addition, module correlation analysis was performed on module eigenvalues and phenotypic trait data, and Pearson correlation was used to calculate the correlation coefficient between phenotypic trait data and gene module eigenvalues, and their correlation heat maps were drawn. OmicShare Tool 2 is used to draw network visualizations of genes within modules.

### 2.6. qRT-PCR of DEGs

To further validate the analysis of the Illumina sequencing data, we selected ten DEGs associated with the leaf angle for qRT-PCR amplification and specific primers were designed

by Primer-BLAST from NCBI (Supplementary Table S2). The actin gene was used as the internal control to normalize the measured gene expression levels [32]. The same cDNA library was used with RNA sequencing in qRT-PCR analysis. qRT-PCR amplification was performed on the quantum studio 5 real-time PCR systems (Thermo Scientific, Waltham, MA, USA) using Premix Pro Taq HS qPCR Kit (SYBR Green) (Accurate,Changsha, China). Moreover, each real time PCR was performed at 20 μL. Reaction volume included cDNA 2 μL, 10 μmol·L$^{-1}$ forward primer 0.8 μL, reverse primer 0.8 μL, Pro Taq HS Premix 10 μL, Rox reference dye 0.4 μL, and ddH$_2$O 6 μL. The amplification parameters were 42 °C for 2 min and 95 °C for 10 min, then 40 cycles of 94 °C for 15 s and 60 °C for 1 min. The products were verified by melting curve analysis. Relative gene expression was analysed by 2$^{-\Delta\Delta CT}$ calculation methods. Three biological replicates and three technical replicates for each sample and each gene were analyzed.

### 2.7. Statistical Analysis of the Leaf Angle and the Cell Length

Data was analyzed using Microsoft Office 2019 and SPSS 19.0 statistical software (SPSS, Inc., Chicago, IL, USA). Duncan's test ($p < 0.05$) was chosen to show significant differences between the treatments.

## 3. Results

### 3.1. Screening of the Optimal Concentration of Exogenous BR

We found that with the increasing of exogenous BR concentration, the leaf angle of maize B73 increased first and then decreased. In particular, the leaf angle reached the largest point when BR concentration on 50 μmol·L$^{-1}$, and when the leaf angle was 41.7°, whereas at 100 μmol·L$^{-1}$, it was significantly reduced, indicating that the maize leaf angle was inhibited at this concentration. (Figure 1). Therefore, 50 μmol·L$^{-1}$ is the optimal concentration for studying the regulatory role of BR on maize leaf angle.

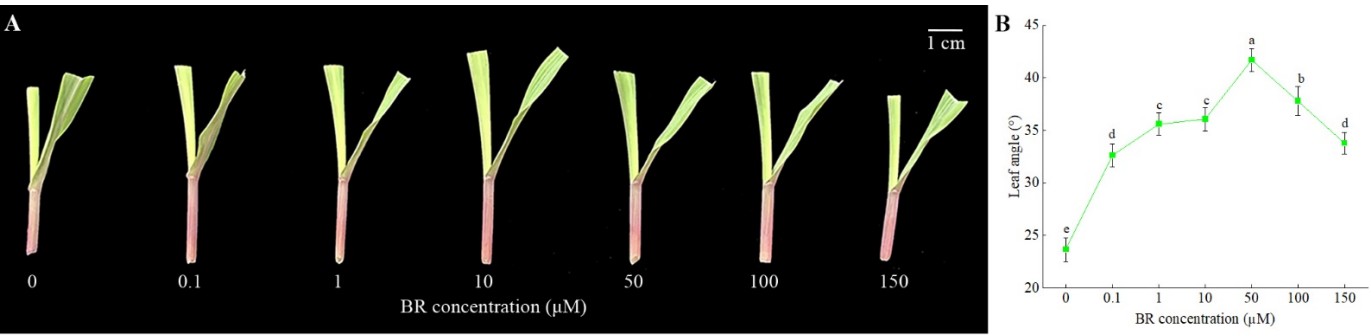

**Figure 1.** Effects of exogenous BR on maize leaf angle. (**A**) Morphologies of maize leaf angle under BR treatment. Scale bars, 1 cm, (**B**) Size of maize leaf angle under BR treatment; different small letters represent significant differences at the 0.05 level.

### 3.2. Cytological Observations of Ligular Region

To further explore the mechanism of exogenous regulation of maize leaf angle, cytological observation was carried out on the cellular structure of maize B73 at the lower edge of the auricle. We found there was a significant difference among these treatments, and the cell length increased first and then decreased with the increase of BR concentration. The cell length was the longest under the treatment of 50 μmol·L$^{-1}$ BR, with an average length of 36.04 μm, an increase of 21.96% compared with the control (Figure 2). Interestingly, the changing trend of cell length near leaf occipital was consistent with the changing trend of leaf angle after different concentrations of BR-treated maize B73.

### 3.3. Analysis of DEGs in B73 under Different Concentrations of BR

To explore mechanistic insights into the regulatory mechanisms of BR on the leaf angle of maize B73, we did RNA sequencing using FPKM as a measure of gene expres-

sion. Gene expression changes were carried out by comparing B73-CK versus B73-1BR, B73-CK versus B73-50BR, and B73-1BR versus B73-50BR. A total of 11,487 DEGs were identified among the three comparisons, and upset plots reflect the distribution of up- and down-regulation of DEGs (Figure 3A). We detected both unique and overlapping sets of DEGs as shown in Figure 3B. Compared with B73-CK, 5328 DEGs (1540 up-regulated and 3788 down-regulated) were identified in B73-1BR (Supplementary Table S3), and 8176 DEGs (3332 up-regulated and 4844 down-regulated) were identified under B73-50BR (Supplementary Table S4). Moreover, 6303 DEGs (3829 up-regulated and 2474 down-regulated) were identified in B73-1BR versus B73-50BR (Supplementary Table S5). Meanwhile, the common distribution of genes in the three comparisons group was 891, of which 156 were up-regulated and 224 were down-regulated (Figure 3B,C). Overall, these shared DEGs reflect the BR-responsive of maize B73 at the transcriptional level, which may aid in clarifying the role of exogenous BR in adjustment of the leaf angle of maize.

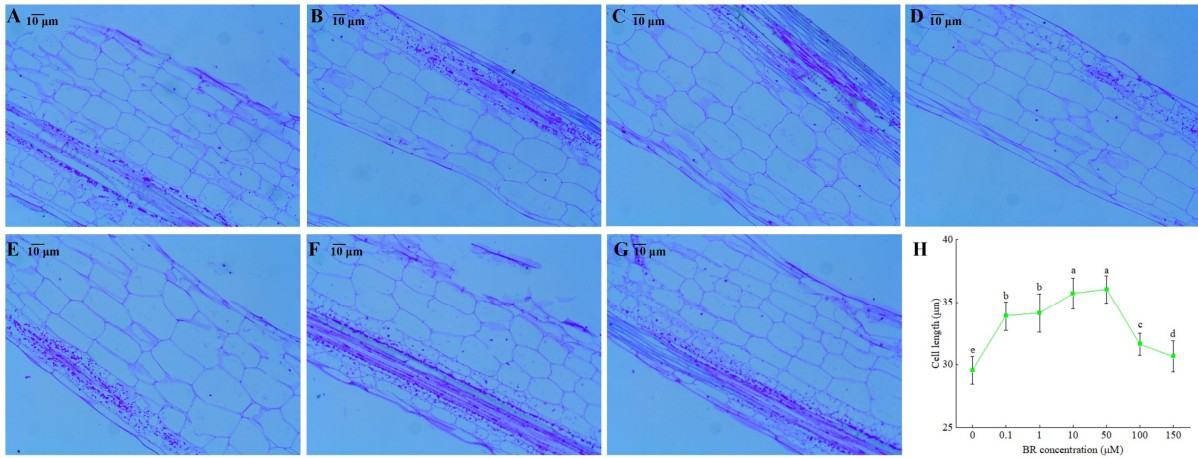

**Figure 2.** The longitudinal structure of leaf auricle-sheath junction of maize under different concentrations exogenous BR. (**A**) 0 $\mu mol \cdot L^{-1}$ BR, (**B**) 0.1 $\mu mol \cdot L^{-1}$ BR, (**C**) 1 $\mu mol \cdot L^{-1}$ BR, (**D**) 10 $\mu mol \cdot L^{-1}$ BR, (**E**) 50 $\mu mol \cdot L^{-1}$ BR, (**F**) 100 $\mu mol \cdot L^{-1}$ BR, (**G**) 150 $\mu mol \cdot L^{-1}$ BR. All bars are 10 $\mu m$, and (**H**) Cell length of the leaf-occipital junction; different small letters represent significant differences at the 0.05 level.

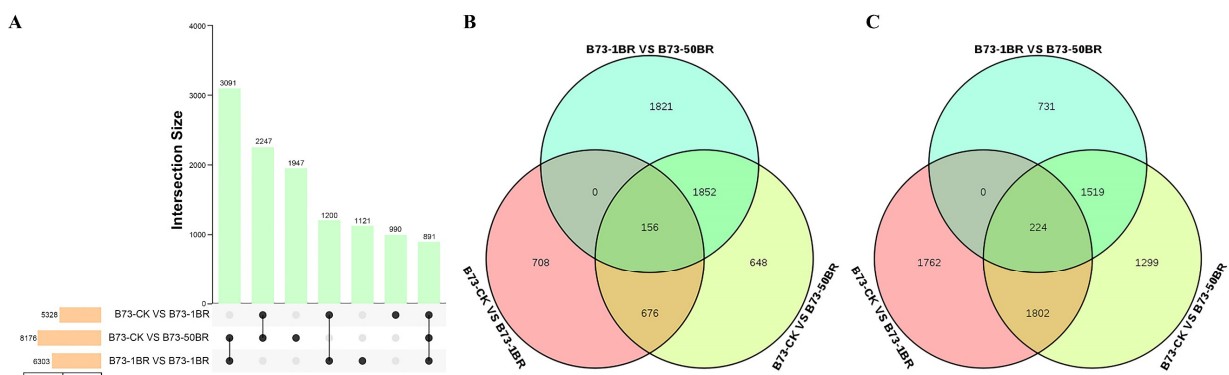

**Figure 3.** Analysis of DEGs in B73 under different concentrations of BR treatment. (**A**) Upset plot analysis of all DEGs in B73 treated with different concentrations of BR, (**B**) Up-regulated DEGs identified in B73 under BR treatment, (**C**) Down-regulated DEGs identified in B73 under BR treatment. The columns indicate DEGs under single or multiple processes, the *x*-axis represents combinations of different groups (black dots represent single-treatment comparisons, black lines connected by dots represent multiple-treatment comparisons), and the *y*-axis represents the number of genes corresponding to the combinations.

### 3.4. Gene Ontology Classification Analyses of B73 DEGs under Different BR Concentrations

To further understand the biological function of DEGs in maize leaf angle under different concentrations of exogenous BR treatment, we carried out the Gene Ontology classification, using the gene ontology (GO) function annotation to analyze all DEGs and divide them into biological processes, cellular components, and molecular functions (Figure 4). The DEGs between material B73-CK versus B73-1BR are mainly enriched in biological processes including carbohydrate metabolic process, cell wall organization or biogenesis and cell wall organization, cellular components include extracellular region, and molecular function as catalytic activity, and hydrolase activity and glucosyltransferase activity (Figure 4A, Supplementary Table S6); DEGs between B73-CK versus B73-50BR are mainly enriched in biological processes including transcription, RNA biosynthetic process and nucleic acid-templated transcription, cellular components include extracellular region, molecular functions include oxidoreductase activity, tetrapyrrole binding, and heme binding (Figure 4B, Supplementary Table S7); DEGs between materials B73-1BR versus B73-50BR are mainly enriched in biological processes aerobic carbohydrate metabolic process, cellular carbohydrate metabolic process, and secondary metabolic process, and cellular components are intrinsic components of membrane, molecular functions are tetrapyrrole binding, and heme binding and oxidoreductase activity (Figure 4C, Supplementary Table S8). Therefore, we speculated that they may play an important role in the formation of maize leaf angle.

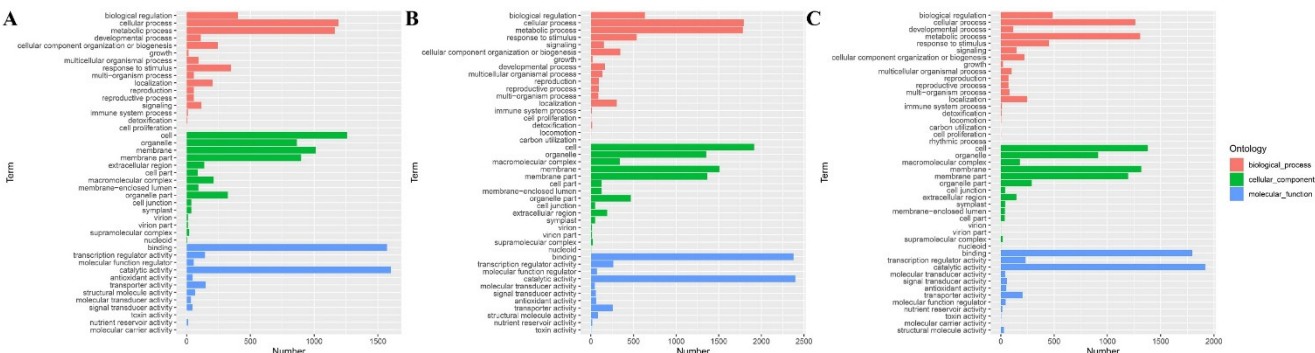

**Figure 4.** GO annotations of DEGs in B73 identified under different treatments. (**A**) B73-CK vs. B73-1BR, (**B**) B73-CK vs. B73-50BR, (**C**) B73-1BR vs. B73-50BR. CK = control; 1 = 1 μmol·L$^{-1}$ BR and 50 = 50 μmol·L$^{-1}$ BR.

### 3.5. Pathway Enrichment Analysis of B73 DEGs under Different BR Concentrations

We performed pathway enrichment analysis on all DEGs, and selected the top 20 metabolic pathways from each comparison (Figure 5). Pathway enrichment almost overlapped under different concentrations of exogenous BR treatment, and significantly enriched pathways comparing B73-CK vs. B73-1BR differences were Arachidonic acid metabolism, Phenylpropanoid biosynthesis, Ubiquinone, and other terpenoid-quinone biosynthesis. The significantly enriched pathway comparing the differences between B73-CK vs. B73-50BR was another glycan degradation. Significantly enriched pathways comparing B73-1BR vs. B73-50BR differences were plant hormone signal transduction, Linoleic acid metabolism, and MAPK signaling pathway-plant.

### 3.6. Gene Co-Expression Network Analysis

We used WGCNA to identify co-expression patterns among DEGs, which clusters gene sets with similar expression patterns into modules. Twenty-one co-expression modules were constructed from the expression data of 9 samples with FPKM values > 1 (Figure 6A,B). We explored the relationship between maize modules and specific leaf angle/cell length under normal and exogenous BR treatment. It has been previously demonstrated that different concentrations of exogenous BR have different effects on maize leaf angle and leaf ear cell size. Therefore, our interest was focused on analyzing the modules in which

the angle size of maize leaves was significantly positively correlated with the plum1 and yellow modules (Figure 6C,D).

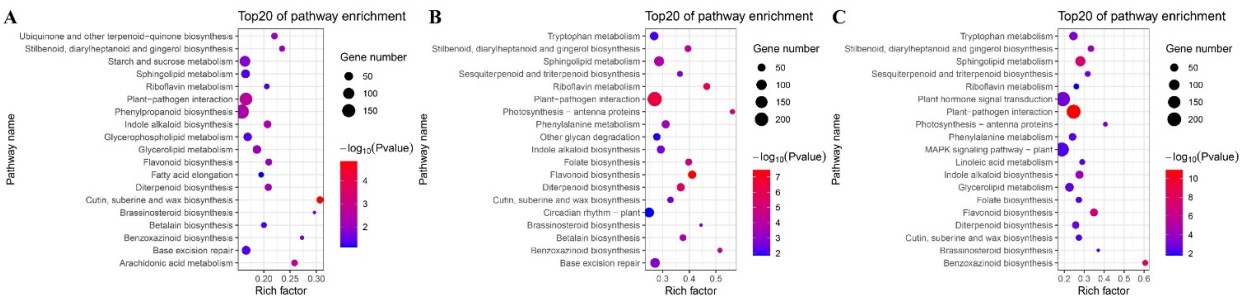

**Figure 5.** Pathway enrichment analysis of B73 exposed to different treatments. (**A**) B73-CK vs. B73-1BR, (**B**) B73-CK vs. B73-50BR, (**C**) B73-1BR vs. B73-50BR. CK = control; 1 = 1 µmol·L$^{-1}$ BR and 50 = 50 µmol·L$^{-1}$ BR.

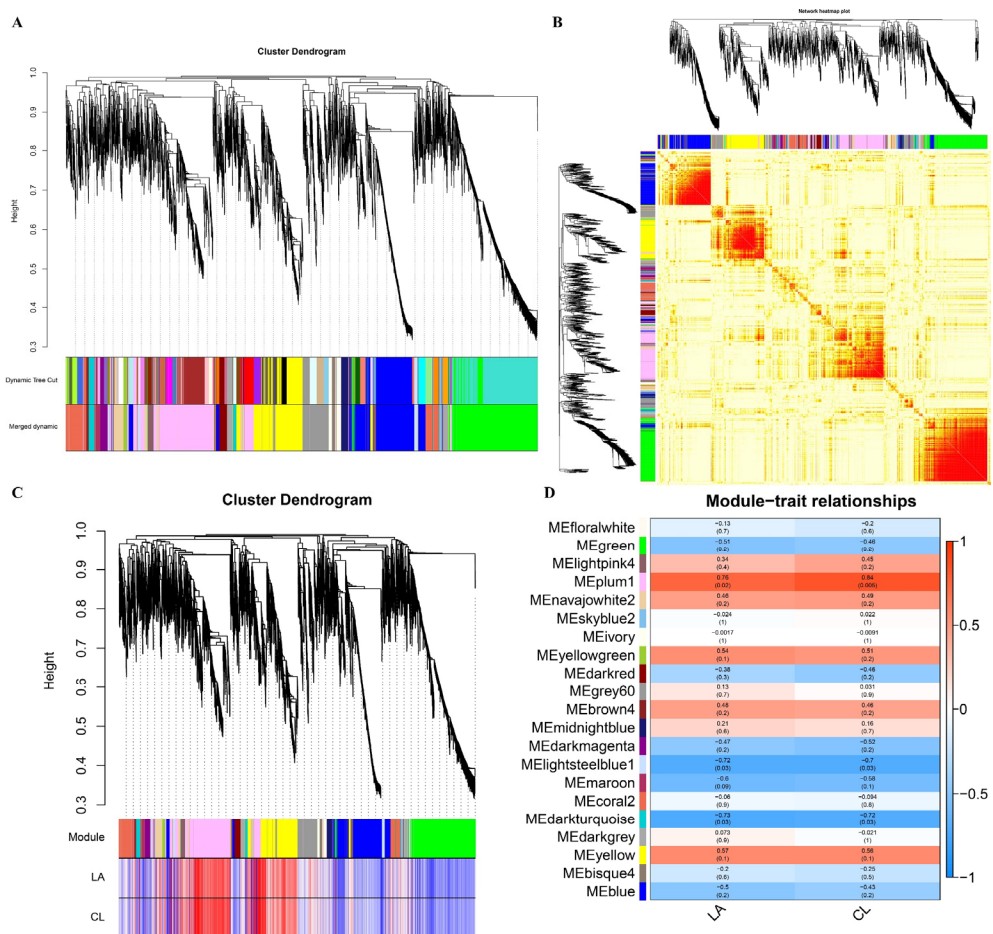

**Figure 6.** Gene cluster analysis and correlation analysis between phenotype and module. (**A**) Hierarchical clustering analysis of co-expression genes, (**B**) Correlated heat maps between modules, (**C**) Correlation between gene module and phenotype, (**D**) Heat map of correlation between gene module and phenotype.

### 3.7. Functional Analysis of Genes in Correlated Modules

To further understand the biological functions of genes on the maize leaf angle related module, we performed a KEGG enrichment analysis. Genes in the plum1 module were mainly involved in Diterpenoid biosynthesis, Stilbenoid, diarylheptanoid, gingerol biosynthesis, Phenylalanine metabolism, Base excision repair, and Brassinosteroid biosynthesis

(Figure 7A). Genes in the yellow module are mainly involved in photosynthesis-antenna proteins, Betalain biosynthesis, Phenylalanine metabolism, Indole alkaloid biosynthesis, and Circadian rhythm-plant (Figure 7B). Thus, we speculated that these pathways may play an important role in regulating the size of the maize leaf angle.

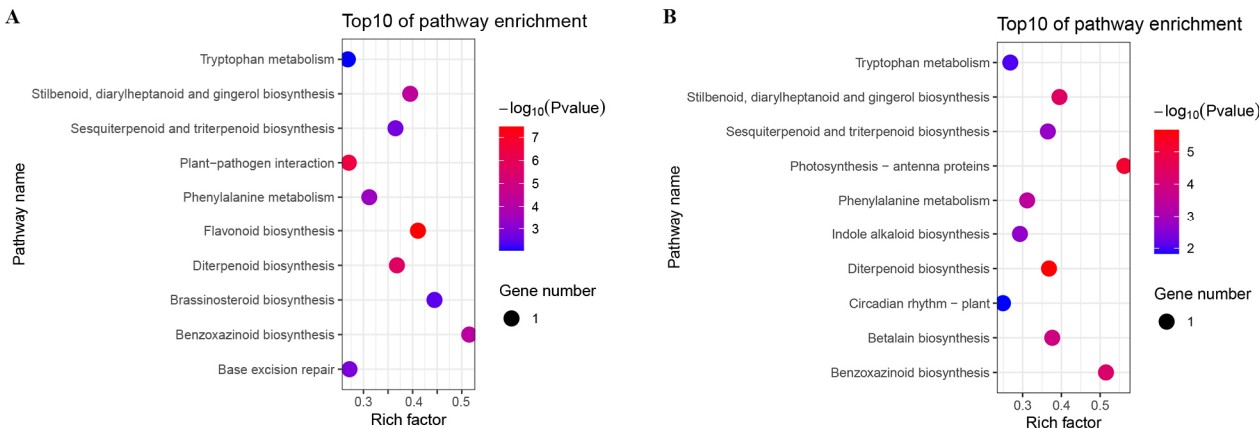

**Figure 7.** Functional analysis of genes in the phenotypic significant enrichment module. (**A**) KEGG enrichment analysis of genes in plum1 module, (**B**) KEGG enrichment analysis of genes in the yellow module.

### 3.8. Analysis of Hub Genes Interaction Network in the Module

In this study, gene network visualization and gene connectivity analysis were performed on the plum1 and yellow module genes. To determine the Hub genes related to the angle of maize leaves, we classified the five genes with the highest kME values in each module as hub genes and utilized the hub genes and their interacting genes to map the gene co-expression network (Figure 8). In the plum1 (0.76) module, the central gene 100279719 was associated with the ribonuclease inhibitor (RI)-like subfamily; 100126972 was associated with the anthocyanin regulatory R-S protein; 103633617 was associated with Phytocyanin; 100272915 was associated with the ubiquitin ligase family; and 103641531 was associated with the Copper amine oxidase (Figure 8A). In the yellow (0.75) module, the central gene 100284778 is related to the ZIM motif family protein; 100193657 is presumed to be related to the MATE efflux family protein; 100285105 is Glucan endo-1,3-beta-glucosidase 13; 100276124 is rho guanine nucleotide exchange factor 18; and 100501519 is alpha, alpha-trehalose-phosphate synthase 5 (Figure 8B). Therefore, we speculate that genes related to these genes are related to the regulation of the maize leaf angle.

### 3.9. Validation of DEGs by qRT-PCR Analysis

To evaluate the reliability of the gene expression profile of maize B73 under normal conditions and after adding different concentrations of exogenous BR, qRT-PCR was used to verify the gene expression level. We selected five genes: Zm00001d013443, Zm00001d053394, Zm00001d026056, Zm00001d013999, and Zm00001d053376 to be up-regulated under the stimulation of different concentrations of exogenous BR, and five genes: Zm00001d043674, Zm00001d003395, Zm00001d022130, Zm00001d044906, and Zm00001d033633 to be down-regulated. We found that the expression levels of these 10 genes in maize B73 at 1 $\mu$mol·L$^{-1}$ and 50 $\mu$mol·L$^{-1}$ BR concentrations were consistent with the transcriptome sequencing results, which indicated that our RNA-seq data were reliable (Figure 9).

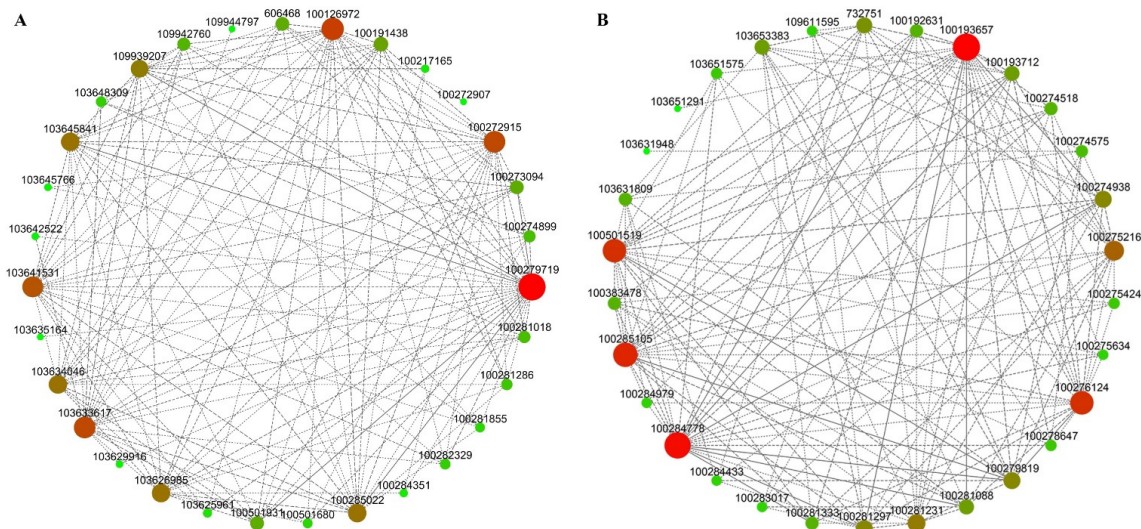

**Figure 8.** Analysis of hub genes network interaction in phenotypic significant enrichment module. (**A**) Network interaction analysis of hub genes in plum1 module, (**B**) Network interaction analysis of hub genes in the yellow module. The size and color gradient of the dots represents the high or low soft threshold of connectivity, with the redder color of the dots representing a higher soft threshold of connectivity.

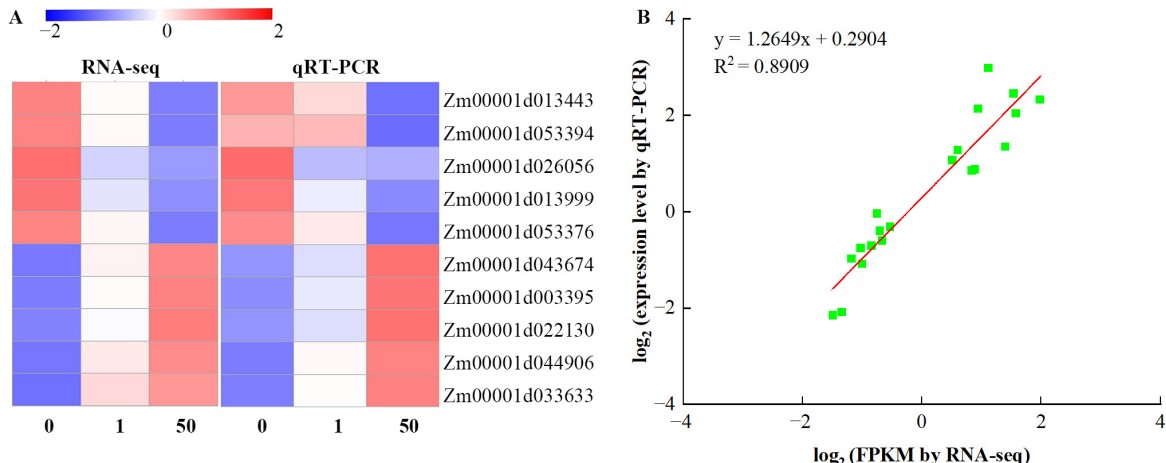

**Figure 9.** The expression pattern of 10 selected genes identified by RNA-seq was verified by qRT-PCR. (**A**) Heat map showing the expression changes (logy-fold change) in response to the B73-CK, B73-1BR, and B73-50BR treatments for each candidate gene as measured by RNA-seq and qRT-PCR, (**B**) Scatter plot showing the changes in the expression (logy-fold change) of selected genes based on RNA-seq via qRT-PCR. Gene expression levels are indicated by coloured bars.

## 4. Discussion

### 4.1. Exogenous BR Promotes the Increase of Maize Leaf Angle

The development of leaf angle in plants is related to many factors, but all of them ultimately affect the size of leaf angle by regulating cell division and cell elongation [33]. BR regulates many important agronomic traits and thus has great potential in agriculture [9]. Previous studies found BR can regulate agronomic traits such as plant height, leaf angle, tillering, and flowering in rice, implying great potential for BR in improving plant performance and productivity [34]. The Epi-rav6 study of the VIVIPAROUS1 (*VP1*) 6 (*RAV6*) gene mutant in rice was found that RAV6 can affect the size of the rice leaf angle by regulating the BR [35]. Studies on rice mutants (*ili1-D*) showed that the leaf angle size could be regulated by brassinolide treatment or over expression of the ILI gene [36]. The

OsBRI1 deletion mutant did not respond to exogenous BR and simultaneously inhibited internal BR signaling, giving the mutant a phenotype with a smaller leaf angle and an upright leaf shape [37]. Previously, many of the proven genes regulating leaf angle are also involved in BR synthesis or hormone signaling pathways, thus demonstrating that BR also plays an important role in the regulation of leaf angle [38]. Consistent with these prior findings, our results showed that different concentrations of exogenous BR to treat maize B73 the leaf angle were significantly increased, and the cell length on the ligular region was also enlarged.

### 4.2. Analysis of DEGs Regulation by Exogenous BR

Transcriptome sequencing can quickly and comprehensively obtain gene expression changes in specific states of different tissues or organs, and differential gene analysis can help to analyze gene regulatory networks [39], and this technology was widely used to excavate functional genes related to different traits [40]. A new study shows there are significant biological metabolic differences in different parts of maize leaves or at different developmental transition states [41]. WGCNA analysis can analyze gene co-expression network through cluster analysis, and combine samples phenotype detection of expression patterns associated with phenotype, to mine key genes associated with phenotype [26]. Studies have shown that leaf occipital plays an important role in the regulation of leaf angle. When the brassinosteroid signal decays in the leaf occipital in rice, it can accelerate the cell division rate of parenchyma tissues at the distal axis, and eventually lead to the upright upward direction of rice leaves [42]. So far, very limited progress has been made in molecular theories on the regulatory mechanism of BR on maize leaf angle. In our study, an appropriate exogenous BR concentration-treated maize B73 increase leaf angle and cell length near the ligular region, as well as BR biosynthesis genes, were induction expression. The DEGs found by transcriptome sequencing were mainly related to plant hormone signal transduction, starch and sucrose metabolism, MAPK signaling pathway-plant, and plant–pathogen interaction.

Plant growth and development is controlled by the action of plant hormones, among which auxin has been implicated in virtually every aspect, such as cell division and cell expansion [43]. In maize, the Teosinte Branched1/CYC loidea/Proliferating Cell Nuclear Antigen Factor (*TCP*) family of transcription factor Branch Angle Defective 1 (TCP) affects the emergence of collateral angles by promoting cell division [44]. The *ZmLPA1* gene is involved in AUX synthesis and response, regulating the maize leaf angle by regulating maize auxin [45]. In our study, we found that the indole-3-acetic acid-amido synthetase gene (Zm00001d043350) and auxin-up RNA 16 (SAUR16) gene (Zm00001d013616) were upregulated in maize B73 under exogenous BR treatment, and that these two genes are correspondingly related to auxin response [42,46]. Moreover, cytokinin (CK) regulates plant growth and morphogenesis. In this process, auxin and brassinoltone are involved in the regulation of cell division and elongation [9]. Arabidopsis-abbreviating CK signaling in the epidermis revealed a contribution of epidermal CK, and CK regulates leaf size by regulating cell division during early leaf development [47]. Here, we found two downregulated genes associated with cytokinin synthesis. The two genes are the cis-zeatin O-glucosyltransferase 2 gene (Zm00001d012335) and cytokinein dehydrogenase gene (Zm00001d032664). Their reduced expression may affect cytokinin synthesis, leading to the reduced cell division speed in maize plants [48].

Starch is the main carbohydrate storage in plants, and sucrose is the primary organic carbon in most higher plants [49]. We found that two genes associated with starch metabolism, the trehalose-6-phosphate synthase (*OtsB*) gene (Zm00001d047110) and α-amylase gene (Zm00001d018159), were up-regulated, and two down-expressed genes were related to sucrose metabolism, the β-fructofuranosidase gene (Zm00001d003776) and β-glucosidase gene (Zm00001d033917). This provided some general information for the gene expression of starch and sucrose metabolism in maize leaf angles under different concentrations of exogenous BR.

A typical MAPK cascade consists of three components: MAPK kinase kinases (MAP-KKKs), MAPK kinases (MAPKKs), and MAPKs. MAPK signaling cascades plays critical roles in diverse processes such as plant growth, development, and defense against pathogens and predators [50,51]. Mutations in the MAPKKK family gene ILA1 led to a decrease in cellulose and the xylan content of the main cell wall components at the occipital site, and abnormal vascular tract development, resulting in reduced mechanical strength at the occipital site and increased leaf angle [46]. Our experimental also found that the three genes related to MAPK signaling: the MAP kinase substrate 1 (*MKS1*) gene (Zm00001d046496), the Ethylene-insensitive3/ethylene-insensitive3-like (*EIN3/EIL*) gene (Zm00001d022530), and the mitogen-activated protein kinase 17/18 (*MAPK17/18*) gene (Zm00001d043741) were up-regulated. We suggested that exogenous BR may enhance MAPK signaling pathways and promote increased maize leaf angle.

DNA replication is not directly linked to the plant cell cycle, and DNA copy numbers can vary widely depending on the tissue and stage of development [52]. A study found that ASYMMETRIC LEAVES1 (*AS1*) and INCURVATA2 (*ICU2*) gene mutants showed increased mRNA levels of the genes for leaf abaxialization [53]. Transcriptome sequencing results showed the downregulation of the DNA polymerase I (*DpoI*) gene (Zm00001d024423) and Arginine methyltransferase-interacting protein gene (GRMZM2G076399) associated with DNA replication. Thus, we hypothesize DNA replication slows during the increasing angle of maize leaves with exogenous BR. By analyzing the results, we found that 3 up-regulated genes were associated with endoplasmic reticulum protein processing and plant-pathogen resistance, and 8 down-regulated genes were associated with ribosome biosynthesis and biotin metabolism (Supplementary Table S6). We speculate that these genes may be associated with maize leaf angle cell enlargement and cell division.

## 5. Conclusions

In this study, we found application of exogenous BR can effectively promote vertical flag leaf formation and cell enlargement at lamina joint position. Based on enrichment of specific functions and metabolic pathways in plants that fared well under BR treatment, we speculate that the mechanism of BR leading to increased leaf angle of maize B73 may be related to these 25 genes (Supplementary Table S9). In addition, we established the molecular model of the mechanisms underlying exogenous BR regulation of leaf angle in the maize V3 stage (Figure 10). Through WGACNA analysis, we identified two modules, plum1 (0.76) and yellow (0.57), that were significantly positively correlated with leaf angle change. Our research provides a theoretical basis for further understanding the complex regulation mechanism of BR on maize leaf angle.

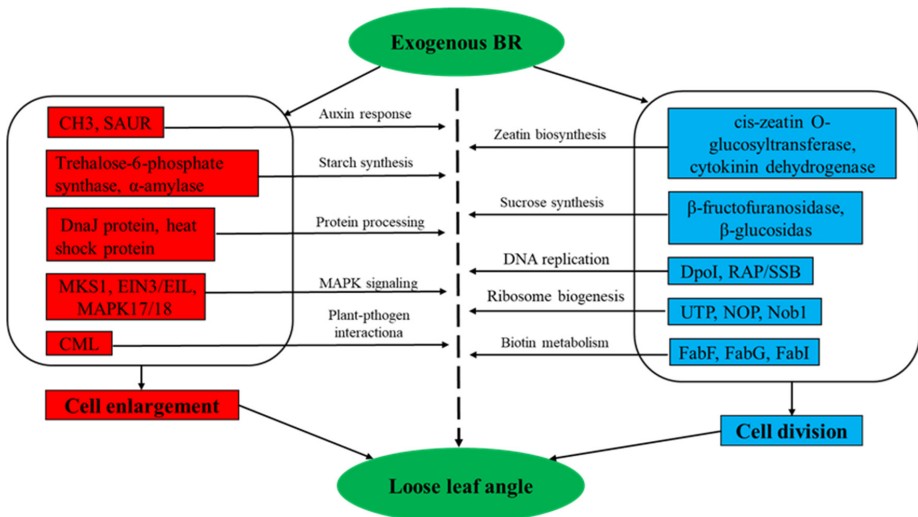

**Figure 10.** Molecular model of the mechanism of maize leaf angle in response to exogenous BR.

**Supplementary Materials:** The following supporting information can be downloaded at: https://www.mdpi.com/article/10.3390/agronomy12081895/s1. Table S1: Summary of the RNA-Seq data analysis; Table S2: Primer sequences used for qRT-PCR analysis in this article; Table S3: DEGs regulated in B73 leaf occipital under 1 μmol·L-1 BR treatment; Table S4: DEGs regulated in B73 leaf occipital under 50 μmol·L-1 BR treatment; Table S5: DEGs regulated in B73 leaf occipital under 1–50 μmol·L-1 BR treatment; Table S6: GO enrichment results of DEGs under 1 μmol·L-1 BR treatment; Table S7: GO enrichment results of DEGs under 50 μmol·L-1 BR treatment; Table S8: GO enrichment results of DEGs under 1–50 μmol·L-1 BR treatment; Table S9: DEGs of maize B73 treated with exogenous BR.

**Author Contributions:** Y.P. designed the experiments. X.J. wrote the manuscript and analyzed data. X.J. and Q.G. performed the experiments. Z.Z., Y.W. and Y.Z. participated in the critical reading and discussion of the manuscript. All authors have read and agreed to the published version of the manuscript.

**Funding:** The industrial support plan for colleges and universities of Gansu, China (No. 2022CYZC-46), the Fuxi Talent Project of Gansu Agricultural University, China (No. GAUFX-02Y09), and the Lanzhou Science and Technology Bureau (No. 2020-RC-122).

**Institutional Review Board Statement:** Not applicable.

**Informed Consent Statement:** Not applicable.

**Data Availability Statement:** Not applicable.

**Acknowledgments:** We thank the reviewers for the critical review of the manuscripts, and Peng Yunling for her guidance and revision.

**Conflicts of Interest:** Authors declared that there is no conflict of interest.

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
