# Peer review of "Transcriptome-Based Weighted Correlation Network Analysis of Maize Leaf Angle Regulation by Exogenous Brassinosteroid"

_agronomy, doi:10.3390/agronomy12081895_

Round 1
Reviewer 1 Report
Dear Authors,
The manuscript submitted for review, entitled "Network analysis of the regulation of maize leaf angle by exogenous BR based on transcriptome and WGCNA" contains a number of shortcomings, errors and omissions.
Two abbreviations appear in the title. I suggest changing the title to "Transcriptome-based weighted correlation network analysis of maize leaf angle regulation by exogenous brassinosteroid".
Numerous linguistic errors appear throughout the manuscript, e.g., sentences without a subject, repetitions, etc. This makes the manuscript very hard to read.
In the introduction, in the second paragraph, the same information is repeated in many sentences. It needs to be rewritten.
Materials and methods.
V3 what is the stage - this needs clarification.
What compound and what solvent was used in the BR analysis? What manufacturer?
What were the sequencing parameters (number of reads/sample, length of reads, paired ends or single ends? Was the RNA enriched in mRNA, or was rRNA removed?
I do not understand the existence of sections 2.4 and 2.5 . Why RNA was isolated by two methods. Moreover, from 2.1 it appears that RNA was isolated by a company. So how?
No description of RT-PCR, reaction composition, primers, what was the reference?
Why is data analysis 2.3 before 2.4 and 2.5? Was an ANOVA performed? and the results? Why was Duncan's test used? What does WT-CK mean, etc.
Results
Figures are mostly barely or completely illegible,so it is difficult to verify the information in the text. Figure 1 lacks a size marker, and Figure 2 it is illegible. What do the error bars mean in these two figures. Were the differences in cell size and leaf angle statistically significant? What are the p values?
Discussion
This part of the manuscript does not actually exist. Section 4.1 is a description of the results. In the other two sections you only refer to six!!!! other papers? The results should be discussed in a much broader context. The literature on this topic is enormous.
Best regards,
M.
Author Response
Response to Reviewer 1 Comments
Dear reviewer:
We appreciate these valuable comments and suggestions very much. We have made detailed corrections in the manuscript corresponding to your suggestions and advice. Thank you for your time and consideration. According to the comments, we have studied the comments carefully and have made a correction which we hope meet with approval. At the same time, we have also made other changes.
Reviewers’ comments are attached below as well as our point-by-point responses shown in red text.
Point 1: The manuscript submitted for review, entitled "Network analysis of the regulation of maize leaf angle by exogenous BR based on transcriptome and WGCNA" contains a number of shortcomings, errors and omissions. Two abbreviations appear in the title. I suggest changing the title to "Transcriptome-based weighted correlation network analysis of maize leaf angle regulation by exogenous brassinosteroid".
Response 1: Thank you for your suggestions. We think your advice is very good. We have revised the title in the manuscript. Line 1-3
Point 2: Numerous linguistic errors appear throughout the manuscript, e.g., sentences without a subject, repetitions, etc. This makes the manuscript very hard to read.
Response 2: Thank you for your suggestions. We have carefully revised the language errors presented in the manuscript.
Point 3: In the introduction, in the second paragraph, the same information is repeated in many sentences. It needs to be rewritten.
Response 3: Thank you for your suggestions. We have reworked the second paragraph in the introduction. Line 41-57.
Materials and methods
Point 4: V3 what is the stage- this needs clarification.
Response 4: V3 stage: The stage when the third leaf of maize is fully expanded, We have explained it in the manuscript. Line 58-60.
Point 5: What compound and what solvent was used in the BR analysis? What manufacturer?
Response 5: The BR (B8780) in the test was produced by Solarbio Company, which was dissolved in 95% ethanol and then prepared into a mother liquor with distilled water. We have explained it in the manuscript. Line 88-90.
Point 6: What were the sequencing parameters (number of reads/sample, length of reads, paired ends or single ends? Was the RNA enriched in mRNA, or was rRNA removed?
Response 6: The library preparations were sequenced on BGISEQ-500 platform and paired-end reads were generated (Line 107-108). The reads length, Clean Reads,Mapped Reads,Mapped Ratio was replenish in the Supplementary Table1 (Line 112-115).。
The RNA library for transcrriptome sequencing, Briefly, mRNA was purified from total RNA using poly-T oligo-attached magnetic beads, Fragmentation was carried out using divalent cations under elevated temperature in NEBNext First Strand Synthesis Reaction Buffer(5X). First strand cDNA was synthesized using random hexamer primer and M-MuLV Reverse Transcriptase(RNase H-). Second strand cDNA synthesis was subsequently performed using DNA Polymerase I and RNase H. Remaining overhangs were converted into blunt ends via exonuclease/polymerase activities. After adenylation of 3’ ends of DNA fragments, NEBNext Adaptor with hairpin loop structure were ligated to prepare for hybridization. In order to select cDNA fragments of preferentially 150~200 bp in length, the library fragments were purified with AMPure XP system (Beckman Coulter, Beverly, USA). Then 3 μl USER Enzyme (NEB, USA) was used with size-selected, adaptor-ligated cDNA at 37°C for 15 min followed by 5 min at 95°C before PCR. Then PCR was performed with Phusion High-Fidelity DNA polymerase, Universal PCR primers and Index (X) Primer. At last, PCR products were purified (AMPure XP system). This step was described by Chen et al, we have added references in Line 105-106
Point 7: I do not understand the existence of sections 2.4 and 2.5 . Why RNA was isolated by two methods. Moreover, from 2.1 it appears that RNA was isolated by a company. So how?
Response 7: We have revised sections 2.4 and 2.5, RNA for transcriptome sequencing and qRT-PCR validation was the same, we have crroected. Line 101-107, Line 139-152.
Point 8: No description of RT-PCR, reaction composition, primers, what was the reference?
Response 8: We described the RT-PCR at section 2.6 and added the required information. Line 139-152
Point 9: Why is data analysis 2.3 before 2.4 and 2.5? Was an ANOVA performed? and the results? Why was Duncan's test used? What does WT-CK mean, etc.
Response 9: We rescaled the position of data analysis 2.3, with the rescaled position at section2.5. and added the Duncan's test results to the manuscript."WT-CK" is a description error that should be "B73-CK," and we have revised such errors in the manuscript.
Point 10: Figures are mostly barely or completely illegible,so it is difficult to verify the information in the text. Figure 1 lacks a size marker, and Figure 2 it is illegible. What do the error bars mean in these two figures. Were the differences in cell size and leaf angle statistically significant? What are the p values?
Response 10: We have again replaced the pictures in the article, The scale bar has added in Figure 1, and reworked Figure 2. The p-Value < 0.05. Line 154-156.
Discussion
Point 11:
This part of the manuscript does not actually exist. Section 4.1 is a description of the results. In the other two sections you only refer to six!!!! other papers? The results should be discussed in a much broader context. The literature on this topic is enormous.
Response 11: We have redescribed the Section 4 and added the relevant papers. See Discussion part.
Thank you again for your detailed and significant suggestions. Based on your comments, we have revised the corresponding content in the manuscript and hope that the correction will meet with your approval.
Best wishes!
Reviewer 2 Report
The manuscript fits within the general scope of the Journal and especially the section on crop breeding. The authors covered a very interesting topic. The introduction is well written the materials and methods section is very detailed and well written also. I have only a minor comment concerning the conclusion section which should not be a summary of the results. An additional effort by the authors should be made to improve this section.
Good luck
Author Response
Response to Reviewer 2 Comments
Dear reviewer:
We appreciate these valuable comments and suggestions very much. We have made detailed corrections in the manuscript corresponding to your suggestions and advice. Thank you for your time and consideration. According to the comments, we have studied the comments carefully and have made a correction which we hope meet with approval. At the same time, we have also made other changes.
Reviewers’ comments are attached below as well as our point-by-point responses shown in red text.
Point 1: It is unclear to the reader which angle is more favorable in Line 31– 34.
Response 1: A smaller leaf angle is better beneficial to the middle leaves to capture light, we have explained in Line 32-34.
Point 2: L47– 49. I suggest explaining the functions of the described proteins..
Response 2: Response 2: Thank you for your suggestions. We have explained the function of the described proteins.(Line 50-54)
Point 3: L52– 53. I suggest a more specific description of the functions of these genes, not just their symbols. This remark also applies to other passages in the introduction.
Response 3: We have added specific description of gene functions in the article.Line 58-60.
Point 4: L74– This section of the manuscript requires major revision. The description of materials and methods is chaotic and incomplete.
Response 4: We have rewritten this part, Line 82-155.
Point 5: L 85– what parameters were measured?
Response 6: Ligular region longitudinal structure were observed and analysis (Line 96-100 ) .
Point 6: L 91– I suggest moving this part (2.3) to the end of the "Materials and Methods" section.
Response 6: We have moved 2.3 Data analysis to 2.9 Statistical Analysis of the Data (Line 152-155).
Point 7: L 93– For which results was the Duncan test applied? These results are not shown in the charts of fig. 1 and 2.
Response 7: We using Duncan test analyzed the significance of leaf angle and cell length of maize B73 under 7 concentrations BR treatment, and we added the significant difference in the Figure 1 and Figure 2.
Point 8: L94– These abbreviations need explanation, I suggest introducing these symbols in part 2.1, after describing the BR treatments.
Response 8: The abbreviations (B73-CK, B73-1BR and B73-50BR) was explanation in the part 2.1. (Line 91-95
Point 9: L107– For which analyzes were RNA and cDNA preparations prepared? There is no description of further procedures. No information from which samples the RNA was isolated, how the samples were collected, from which experimental combinations, how many repeats, etc.
Response 9: We extracted total RNA and constructed the cDNA library for RNA-Seq and qRT-PCR, the experimental method see the Line 91-95, Line 101-108, Line 138-152 for details.
Point 10: L113– RNA extraction has already been described in the previous paragraph (2.4). The description covers a different procedure than the one presented in the section 2.5. No information from which samples the RNA was isolated, from which experimental combinations, how many repeats, etc.
Response 10: This part was rewritten, see the details at 2.3 (Line 101-108) and 2.6 (Line 138-152) part.
Point 11: L115– from which experimental combinations the samples came from, in how many repetitions?
Response 11: The samples came from B73-CK, B73-1BR and B73-50BR treatment, each treatment have 3 biological samples. Line 102-103, Line 142-143.
Point 12: L118– 120. The sentence is unclear.
Response 12: This description was the experimental method about ligular region longitudinal structure observation, we put this sentence in 2.2 part, Line 96-100
Point 13: L120– This is described previously.
Response 13: We have deleted this sentence.
Point 14: L129– previously stated that 7 combinations of concentrations were tested
Response 14: In our study, we used 7 BR concentrations treated maize seedlings at 3rd-leaf period of maize B73, we select two concentration (1 and 50 mmol ·L-1) for RNA-Seq after analysis the leaf angle and cell length. Here, we state error, have revise it in the manuscript Line 128-129.
Point 15: Figure 2. No description in the photos. I suggest indicating the individual structures on the pictures. The inscriptions on the pictures are impossible to read.
Response 15: We have added description of the information in graph title.
Point 16: L164– it is not clear which picture relates to which experimental combination
Response 16: We have explained it in detail in Line 181-184.
Point 17: L178 – 180. The sentence is not clear, what was compared in theis analysis? The numbers don't match.
Response 17: This sentence describing error, we have corrected it at Line 196-198.
Point 18: Figure 3. The descriptions on the pictures are impossible to read. The reader is unable to obtain any information.
Response 18: We redrawn (Figure 3), to make it more readable.
Point 19: Figure 4. The descriptions on the pictures are impossible to read. The reader is unable to obtain any information.
Response 19: We have redrawn Figure 4 and it was readable.
Point 20: Figure 5. The descriptions on the pictures are impossible to read. The reader is unable to obtain any information.
Response 20: We have provided a high-definition of Figure 5 make it more readable.
Thank you again for your detailed and significant suggestions. Based on your comments, we have revised the corresponding content in the manuscript and hope that the correction will meet with your approval.
Best wishes!

Reviewer 3 Report
The manuscript describes an interesting experiment. A lot of work was done and many results were analysed. However, the manuscript is problematic in many ways, it is poorly written and needs major changes to be accepted.
General comments
1. The introduction is quite poor, more information is needed on BR and their roles in cell growth and division.
2. The description of materials and methods is quite chaotic and not precise, a lot of information is missing - should be greatly improved.
3. Abbreviations should be explained. We do not receive descriptions of gene functions, the authors only use their symbols.
4. The quality of some figures is poor, descriptions are very small and unreadable.
5. The captions of some figures are insufficient or incomplete.
6. The part “Discussion” is very poor, this section repeats the methods and results. The same remarks apply to the “Conclusions”. I propose to provide more detailed information about the identified genes, pathways and processes and the possible role of the identified DEGs in regulating cell growth and division.
7. The language needs to be improved. Attention should be paid to more precise formulation of sentences so that the correct meaning is well understood by the readers.
Detailed comments are marked on the PDF version of the manuscript.

Author Response
Response to Reviewer 3 Comments
Dear reviewer:
We appreciate these valuable comments and suggestions very much. We have made detailed corrections in the manuscript corresponding to your suggestions and advice. Thank you for your time and consideration. According to the comments, we have studied the comments carefully and have made a correction which we hope meet with approval. At the same time, we have also made other changes.
Reviewers’ comments are attached below as well as our point-by-point responses shown in red text.
General comments
- The introduction is quite poor, more information is needed on BR and their roles in cell growth and division.
Response 1: We have information about BR roles in cell growth and division, modifications have been made and written in red in the text.
- The description of materials and methods is quite chaotic and not precise, a lot of information is missing - should be greatly improved.
Response 2: The description of materials and methods were rewritten (Line 82-155).
- Abbreviations should be explained. We do not receive descriptions of gene functions, the authors only use their symbols.
Response 3: All the abbreviations (gene or protein) in the manuscript were explained, and marked in red.
- The quality of some figures is poor, descriptions are very small and unreadable.
Response 4: Figure 1-10 was redrawn, and we found the default picture resolution in word is 96dpi, if the resolution of the picture we insert is not 96dpi, the picture will be deformed. We have set the picture resolution was high-fidelity.
- The captions of some figures are insufficient or incomplete.
Response 5: We have add information of Figure 2 (A~G).
- The part “Discussion” is very poor, this section repeats the methods and results. The same remarks apply to the “Conclusions”. I propose to provide more detailed information about the identified genes, pathways and processes and the possible role of the identified DEGs in regulating cell growth and division.
Response 6: The Discussion and Conclusions part was rewritten, added some information about identified DEGs in regulating cell growth and division aspect, the detail in “Discussion Part”
- The language needs to be improved. Attention should be paid to more precise formulation of sentences so that the correct meaning is well understood by the readers.
Response 7: Ambiguous syntax, inaccurate grammar, spelling, and punctuation were revised.
Detailed comments marked on the PDF version of the manuscript.
Point 1: Line 31– 34. It is unclear to the reader which angle is more favorable. Response 1: A smaller leaf angle is better beneficial to the middle leaves to capture light, we have explained in Line 32-34.
Point 2: L47– 49. I suggest explaining the functions of the described proteins.
Response 2: Thank you for your suggestions. We have explained the function of the described proteins.(Line 50-54)
Point 3: L52– 53. I suggest a more specific description of the functions of these genes, not just their symbols. This remark also applies to other passages in the introduction.
Response 3: We have added specific description of gene functions in the article.Line 58-60.
Point 4: L74– This section of the manuscript requires major revision. The description of materials and methods is chaotic and incomplete.
Response 4: We have rewritten this part, Line 82-155.
Point 5: L 83–84. of which experimental combinations the RNA-seq analysis was performed, in how many replicates?
Response 5: RNA-seq tissue sample were 2 cm in the lobe occipital region under 0, 1 and 50 μmol·L-1 treatment, each treatment collect 3 samples, total of 9 samples were collected. This was clarified in Line 91-95.
Point 6: L 85– what parameters were measured?
Response 6: Ligular region longitudinal structure were observed and analysis (Line 96-100 )
Point 7: L 91– I suggest moving this part (2.3) to the end of the "Materials and Methods" section.
Response 7: We have moved 2.3 Data analysis to 2.9 Statistical Analysis of the Data (Line 152-155).
Point 8: L 93– For which results was the Duncan test applied? These results are not shown in the charts of fig. 1 and 2.
Response 8: We using Duncan test analyzed the significance of leaf angle and cell length of maize B73 under 7 concentrations BR treatment, and we added the significant difference in the Figure 1 and Figure 2.
Point 9: L94– These abbreviations need explanation, I suggest introducing these symbols in part 2.1, after describing the BR treatments.
Response 9: The abbreviations (B73-CK, B73-1BR and B73-50BR) was explanation in the part 2.1. (Line 91-95)
Point 10: L107– For which analyzes were RNA and cDNA preparations prepared? There is no description of further procedures. No information from which samples the RNA was isolated, how the samples were collected, from which experimental combinations, how many repeats, etc.
Response 10: We extracted total RNA and constructed the cDNA library for RNA-Seq and qRT-PCR, the experimental method see the Line 91-95, Line 101-108, Line 138-152 for details.
Point 11: L113– RNA extraction has already been described in the previous paragraph (2.4). The description covers a different procedure than the one presented in the section 2.5. No information from which samples the RNA was isolated, from which experimental combinations, how many repeats, etc.
Response 11: This part was rewritten, see the details at 2.3 (Line 101-108) and 2.6 (Line 138-152) part.
Point 12: L115– from which experimental combinations the samples came from, in how many repetitions?
Response 12: The samples came from B73-CK, B73-1BR and B73-50BR treatment, each treatment have 3 biological samples. Line 102-103, Line 142-143.
Point 13: L118– 120. The sentence is unclear.
Response 13: This description was the experimental method about ligular region longitudinal structure observation, we put this sentence in 2.2 part, Line 96-100.
Point 14: L120– This is described previously.
Response 14: We have deleted this sentence.
Point 15: L129– previously stated that 7 combinations of concentrations were tested
Response 15: In our study, we used 7 BR concentrations treated maize seedlings at 3rd-leaf period of maize B73, we select two concentration (1 and 50 mmol ·L-1) for RNA-Seq after analysis the leaf angle and cell length. Here, we state error, have revise it in the manuscript Line 128-129.
Point 16: Figure 2. No description in the photos. I suggest indicating the individual structures on the pictures. The inscriptions on the pictures are impossible to read.
Response 16: We have added description of the information in graph title.
Point 17: L164– it is not clear which picture relates to which experimental combination
Response 17: We have explained it in detail in Line 181-184.
Point 18: L178 – 180. The sentence is not clear, what was compared in theis analysis? The numbers don't match.
Response 18: This sentence describing error, we have corrected it at Line 196-198.
Point 19: Figure 3. The descriptions on the pictures are impossible to read. The reader is unable to obtain any information.
Response 19: We redrawn (Figure 3), to make it more readable.
Point 20: L191. This part requires improvement, a wider presentation of individual groups of identified genes, the more that the charts are unreadable.
Response 20: We have re-described this part at Line 209-227, and Figure 4 provided a high-definition picture.
Point 21: Figure 4. The descriptions on the pictures are impossible to read. The reader is unable to obtain any information.
Response 21: We have redrawn Figure 4 and it was readable.
Point 22: Figure 5. The descriptions on the pictures are impossible to read. The reader is unable to obtain any information.
Response 22: We have provided a high-definition of Figure 5 make it more readable.
Point 23: L274–280. This fragment belongs rather to the "Materials and Methods". The functions of the analyzed genes should be explained.
Response 23: We have added theses genes functions in Supplemental Table 1.
Point 24: "Discussion" needs drastic changes.
Response 24: We have modified "Discussion" part.
Point 25: L297–307. This information has already been provided before, it is not a discussion of the results.
Response 25: We have modified "Discussion" part.
Point 26: L318–328. These are the results, not the discussion.
Response 26: We have modified "Discussion" part.
Point 27: L330–348. These are the results not the discussion. The functions and presumed role of the genes mentioned should be stated, not the symbols themselves.
Response 27: We have modified "Discussion" part.
Point 28: L352–366. "Conclusions" should be changed. The description of the experiment and some results is repeated again and again, not the conclusions.
Response 28: We re-describe "Conclusions" in the manuscript.
Thank you again for your detailed and significant suggestions. Based on your comments, we have revised the corresponding content in the manuscript and hope that the correction will meet with your approval.
Best wishes!
Round 2
Reviewer 1 Report
Dear Authors,
The manuscript is definitely better after the revisions; it has only a few minor inadequacies already.
Line 32: replace "better beneficial" with profitable
Line 78 and 79 - clarify the abbreviations V3 and WGCNA
Line 89 - solvent not known, although the review response states that solvent information was added.
Fig.1 Not sure what the scale is. The size marker requires the unit, it reflects
Fig. 2 the size marker is still not very visible, and it is also unclear what unit and what value it represents.
Best regards,
M.
Author Response
Response to Reviewer 1 Comments
Dear reviewer:
We appreciate these valuable comments and suggestions very much. We have made detailed corrections in the manuscript corresponding to your suggestions and advice. Thank you for your time and consideration. According to the comments, we have studied the comments carefully and have made a correction which we hope meet with approval. At the same time, we have also made other changes.
Reviewers’ comments are attached below as well as our point-by-point responses shown in red text.
Point 1: Line 32: replace "better beneficial" with profitable
Response 1: Thank you for your suggestions. We have replaced "better beneficial" with "profitable" in Line32.
Point 2: Line 78 and 79 - clarify the abbreviations V3 and WGCNA
Response 2: Thank you for your suggestions. We have clarified the abbreviations V3 at Line83, Line23 clarified the abbreviations WGCNA.
Point 3: Line 89 - solvent not known, although the review response states that solvent information was added.
Response 3: Thank you for your suggestions. We have added solvent information to Line95-96.
Point 4: Fig.1 Not sure what the scale is. The size marker requires the unit, it reflects
Response 4: We have added the scale bar’s size make in the Figure 1.
Point 5: Fig. 2 the size marker is still not very visible, and it is also unclear what unit and what value it represents.
Response 5: We have added the size marker and the unit.
Thank you again for your detailed and significant suggestions. Based on your comments, we have revised the corresponding content in the manuscript and hope that the correction will meet with your approval.
Best wishes!

Reviewer 3 Report
The manuscript has been largely revised.
General comments
1. The title is much better than the previous one.
2. The introduction does not seem to be completely adequate. There is a lack of information on the interaction of BRs with auxins and cytokinins, which is detailed in the discussion as an important aspect in the regulation of cell growth and division.
2. The description of materials and methods has been improved. Some shortcomings still require improvement.
4. The figures were improved. Fig. 1 and 2 require some correction.
6. The part Discussion has been improved. Unfortunately, I still have the impression that the Introduction, Results and Discussion are not logically related. The same remarks apply to the Conclusions. I propose to rethink the results obtained and to draw strict conclusions as well as to provide the suitable background in the introduction.
7. The language still needs some improvement.
Detailed comments are marked on the PDF version of the manuscript.

Author Response
Response to Reviewer 3 Comments
Dear reviewer:
We appreciate these valuable comments and suggestions very much. We have made detailed corrections in the manuscript corresponding to your suggestions and advice. Thank you for your time and consideration. According to the comments, we have studied the comments carefully and have made a correction which we hope meet with approval. At the same time, we have also made other changes.
Reviewers’ comments are attached below as well as our point-by-point responses shown in red text.
General comments
- The introduction does not seem to be completely adequate. There is a lack of information on the interaction of BRs with auxins and cytokinins, which is detailed in the discussion as an important aspect in the regulation of cell growth and division.
Response 1: We added information about BR and auxins regulatory plant growth and cell division. Line 47-52.
- The description of materials and methods has been improved. Some shortcomings still require improvement.
Response 2: We have improved this part according the detailed comments marked on the PDF version.
- The figures were improved. Fig. 1 and 2 require some correction.
Response 3: The figure 1 and 2 have improved.
- The part Discussion has been improved. Unfortunately, I still have the impression that the Introduction, Results and Discussion are not logically related. The same remarks apply to the Conclusions. I propose to rethink the results obtained and to draw strict conclusions as well as to provide the suitable background in the introduction.
Response 4: We have reworked the Discussion and Results based on the results and added appropriate context to the Introduction.
- The language still needs some improvement.
Response 5: Thank you for your valuable and thoughtful comments. We have carefully checked and improved the English writing in the revised manuscript.
Detailed comments marked on the PDF version of the manuscript.
Point 1: L23– I think the abbreviation should be clarified.
Response 1: We have clarified the abbreviation the WGCAN at Line 23.
Point 2: L118–119. Differentially expressed genes
Response 2: ‘Differential expression genes’ have changed to ‘Differentially expressed genes’, Line 124.
Point 3: L139–140. I do not fully understand what was the criterion for selecting genes for the analysis, what means "significance"? Is it that the expression of these genes differed significantly between the different experimental combinations? It seems to me that a better criterion would be to select DEGs which, due to their function, may be related to the process under study (leaf angle).
Response 3: we have modified this sentence to ‘ we selected ten DEGs associated with the leaf angle for qRT-PCR amplifications’ ,Line 145-146.
Point 4: L143–144. Data on plant material and the method of obtaining RNA / cDNA are needed. From the text I suppose it was the same cDNA that was used in RNA-seq but the sentence is unclear.
Response 4: We have modigied ‘ The cDNA library for this purpose was followed as a transcriptome analysis described in above’ to ‘The same cDNA library were used with RNA sequencing in qRT-PCR analysis.’ Line 148-149.
Point 5: L146– the same is repeated at the end of the paragraph.
Response 5: We have deleted the sentence ‘Each treatment had three biological replicates’.
Point 6: L148– concentration? Only forward primer?
Response 6: We have added the concentration and reverse primer, the sentence was ‘10 μmol·L-1 forward primer 0.8 μl , reverse primer 0.8 μl’. Line 153.
Point 7: L153– I propose to specify which results were statistically analyzed.
Response 7: ‘Statistical Analysis of the Data’ have revised to ‘‘Statistical Analysis of the leaf angle and cell length’, Line 159.
Point 8: L160–162. this information has already been provided in previous sections of the manuscript, here we only list the results
Response 8: We have deleted this information.
Point 9: L 170–I propose to explain what the letters (a, b etc.) mean in the chart
Response 9: We have explained the small letter in the chart, ‘different small letters represent significant differences at 0.05 level.
Point 10: L181– it is necessary to indicate (with arrow) and sign of the individual tissues of the leaf on the pictures.
Response 10: We have added the size marker and the unit in figure 2. We aim to illustrate the change in cell size of leaf auricle - sheath junction through Figure 2. By consulting a lot of literature about cell morphology, study several cell structures changes need to be labelled tissues,in our study we observation only one type of cell, and it is similar to the study of Chen et al. (https://doi.org/10.3389/fpls.2021.739101).
Point 11: L183–184. clarify the meaning of the letters.
Response 11: We have explained the small letter in the chart, ‘different small letters represent significant differences at 0.05 level.
Point 12: L198–199. This sentence needs to be corrected. The identified DEGs are associated with various biological processes that are influenced by BR.Certainly, the effect of the treatment of maize with BR was not only limited to the change in the angle of the leaves, but also to other processes that were not analysed in this work. The work simply identified genes that were differentially expressed as a result of BR maize treatment. The role of the authors is to try to explain which of these genes may be related to the studied process.
Response 12: The sentence‘ These shared DEGs may play an important role in the adjustment of leaf angle in maize in response to exogenous BR’ has replace with ‘Overall, these shared DEGs reflecting the BR-responsive of maize B73 at transcriptional level, which may aid in clarifying the role of exogenous BR in adjustment of leaf angle of maize. ’, Line 204-206.
Point 13: L226–227. as I wrote before, these results apply to all changes induced by BR treatment
Response 13: The sentence ‘The results showed that the Gene Ontology of maize regulating leaf angle was very different under different concentrations of exogenous BR treatment.’ was revised to ‘Therefore, we speculated that they may play an important role in the formation of maize leaf angle.’, Line 233-234.
Point 14: L237– BRs also have an impact on the processes related to the stress response, hence probably changes in these pathways.
Response 14: We did not find pathway related to stress response pathway form KEGG analysis of B73-CK vs B73-1BR.
Point 15: L363–365. same sentence as above.
Response 15: We have deleted this sentence.
Point 16: L409–412. I propose to make these conclusions even more precise, first listing the processes and then the possible genes responsible for them.
Response 16: We have described this part of the conclusion more precisely. Line 412-416.
Thank you again for your detailed and significant suggestions. Based on your comments, we have revised the corresponding content in the manuscript and hope that the correction will meet with your approval.
Best wishes!
